# Service Function Chain Deployment Method Based on Traffic Prediction and Adaptive Virtual Network Function Scaling

Haiyan Hu [1] , Qiaoyan Kang [1,*], Shuo Zhao [1], Jianfeng Wang [1] and Youbin Fu [2]

[1] College of Information and Navigation, Air Force Engineering University, Xi'an 710077, China
[2] Unit 93107 of People's Liberation Army, Shenyang 110141, China
*   Correspondence: pinky8012@126.com

**Abstract:** With the development of network function virtualization (NFV), the resource management of service function chains (SFC) in the virtualized environment has gradually become a research hotspot. Usually, users hope that they can get the network services they want anytime and anywhere. The network service requests are dynamic and real-time, which requires that the SFC in the NFV environment can also meet the dynamically changing network service requests. In this regard, this paper proposes an SFC deployment method based on traffic prediction and adaptive virtual network function (VNF) scaling. Firstly, an improved network traffic prediction method is proposed to improve its prediction accuracy for dynamically changing network traffic. Secondly, the predicted traffic data is processed for the subsequent scaling of the VNF. Finally, an adaptive VNF scaling method is designed for the purpose of dynamic management of network virtual resources. The experimental results show that the method proposed in this paper can manage the network resources in dynamic scenarios. It can effectively improve the availability of network services, reduce the operating overhead and achieve a good optimization effect.

**Keywords:** deployment of SFCs; traffic forecast; adaptive VNF scaling

## 1. Introduction

For network operators, the management of network resources is a severe task. In daily life, network traffic provided by operators will pass through multiple different network functions in turn to meet customer needs. However, the network traffic changes dynamically. Traditional network functions need to meet the demand of traffic peaks, and the management flexibility is poor, resulting in low resource utilization. The emergence of virtual network functions (VNF) enables operators to flexibly operate network functions and dynamically scale VNF, thereby realizing the dynamic deployment of service function chains (SFC) to improve the utilization of network resources.

The dynamic scaling of VNF mainly includes horizontal scaling (creating or deleting VNF) and vertical scaling (increasing or reducing VNF processing capacity). This paper only considers the horizontal scaling of VNFs. After the scaling is completed, the obtained results are used as the constraints. The SFC deployment problem is modeled as an integer linear programming model. Then we find the shortest path and complete the mapping of the virtual link, which can realize the deployment of SFC in dynamic scenarios. Different from the offline SFC deployment problem [1,2], the dynamic deployment of SFC considers the dynamic change of traffic in dynamic scenarios so that the VNF instance can change according to the change in network traffic. It can realize the dynamic management of network virtual resources. So, achieving effective dynamic scaling of VNF and dynamically deploying SFC has become a major challenge for network resource management [3].

Aiming at the dynamic scaling of VNFs, reference [4] proposes a traffic prediction method based on the traffic characteristics of the operator's network and designs a VNF deployment algorithm based on the prediction results to realize the dynamic scaling

of VNFs. However, the prediction will produce errors. For the problem of inaccurate prediction, reference [5] adopts an online learning method to reduce the error of service chain demand prediction. Moreover, when deploying new VNFs according to the prediction results, it adopts an adaptive scaling strategy to achieve the goal of saving resources and reducing deployment costs. Reference [6] proposes a VNF lifecycle management and online deployment method for the dynamic scaling of VNF and dynamic deployment of SFC, according to the prediction results of network traffic. As a result, the scaling overhead of VNFs and the deployment bandwidth of SFCs in dynamic scenarios are reduced. Reference [7] proposes a traffic-aware algorithm to predict future load states. It proposes a dynamic VNF deployment method based on the prediction results to improve resource utilization. To sum up, network traffic prediction is the primary issue in the study of dynamic scaling of VNFs and the dynamic deployment of SFCs.

Aiming at the problem of network traffic prediction, reference [6] adopts a method based on the GRU neural network to predict network traffic. Reference [7] adopts an algorithm based on the LSTM network to realize traffic awareness. However, these methods based on the neural network often require a large number of data samples and are prone to fall into local optimum, resulting in unstable training results. Relatively speaking, machine learning methods such as the support vector machine (SVM) algorithm have strong generalization ability and global optimality. It can collect and analyze the dependencies of network traffic to achieve the effect of predicting the future network [8]. It works for small samples. References [9,10] use genetic algorithms (GA) and particle swarm optimization (PSO) to optimize the network traffic prediction model based on SVM and realize the chaotic prediction of network traffic. However, traditional GA and PSO algorithms are prone to falling into local optimums. When dealing with high-dimensional problems, the convergence speed is slow, and it is easy to diverge. Reference [11] combined the beetle antennae search (BAS) algorithm with the PSO algorithm and proposed the beetle swarm optimization (BSO) algorithm. The group optimization performance is improved through the principle of beetle foraging, to realize the processing of high-dimensional, complex optimization problems. The experimental results show that the performance of the BSO algorithm is better than that of the PSO and GA algorithms. Moreover, the BSO algorithm is widely used. For example, reference [12] uses the BSO algorithm to improve the convergence speed of the BP neural network. Reference [13] applies the BSO algorithm to the path planning problem. Reference [14] applies the improved BSO algorithm to the personal credit evaluation based on SVM, which effectively improved the classification performance of SVM. Reference [15] proposes an adaptive mutation BSO algorithm, which effectively improved the optimization effect and convergence speed.

It can be seen from the above analysis that the BSO algorithm is widely used in optimization problems, which can effectively improve the convergence speed and optimization accuracy of the algorithm. This provides a new idea for optimizing the parameters of SVM [12]. In this paper, the BSO algorithm is applied to the SVM parameter optimization problem in network traffic prediction. We propose a network traffic chaos prediction model based on the BSO algorithm optimized SVM. However, if the predicted traffic is applied to the dynamic scaling of VNF, the underestimated predicted traffic should be reduced to improve the availability of network services. At the same time, the predicted traffic should not be overestimated to save resource consumption. Therefore, how to predict the upper bound of network traffic is also one of the issues that we need to consider when conducting VNF adaptive scaling [4].

In addition, new problems will arise when scaling the VNF instance. When the number of VNFs required in the future is greater than the number of working VNFs, more VNF instances need to be deployed. This operation consumes a lot of resources and time, resulting in a lot of overhead [5]. When the number of VNFs required in the future is less than the number of working VNFs, if the VNF instance is not deleted in time, a large amount of running overhead will be incurred. Therefore, the design of an appropriate VNF

scaling method to achieve the purpose of reducing operational overhead is another issue that we need to consider.

In summary, this paper proposes an SFC deployment method based on traffic prediction and VNF scaling in the scenario of dynamic traffic changes. Firstly, to achieve adaptive scaling of VNF, we need to predict the upcoming traffic demand. In this regard, we propose a network traffic chaos prediction model based on the improved BSO algorithm optimized for the SVM. It can improve the accuracy of network traffic prediction. Secondly, to apply the predicted traffic to VNF dynamic scaling, we need to minimize the traffic upper limit on the predicted traffic data. This processing can ensure that the traffic data is as high as possible compared to the actual traffic. Thus, we can improve the availability of network services to facilitate the adaptive scaling of VNFs. Then we design a VNF adaptive scaling method based on the processed traffic data. It can flexibly create and delete VNFs to dynamically adjust the number of VNF instances. Therefore, we can realize dynamic management of virtual resources and reduce network operation overhead. Finally, according to the results of the adaptive scaling VNF, we use the k-shortest path algorithm to calculate the service function path. We complete the mapping of virtual links and implement SFC deployment in dynamic scenarios.

The specific contributions are as follows:

1. We propose a network traffic chaos prediction model based on the improved BSO algorithm optimized for the SVM.
2. We process the predicted traffic data to minimize the traffic cap.
3. We design an adaptive VNF scaling method.

The rest of the paper is organized as follows. Section 2 provides a brief description of the problem studied in this paper. In Section 3, we build the corresponding models. In Section 4, we describe the corresponding algorithms. Section 5 is the analysis of the corresponding experimental results. Finally, conclusions are drawn in Section 6.

## 2. Problem Description

### 2.1. Dynamic Deployment of SFC

Figure 1 is a schematic diagram of SFC dynamic deployment, in which Figure 1a is the original SFC deployment method. User 1 enters the network from switch 4 and requests network services for VNF1 and VNF2. Therefore, VNF1 and VNF2 are deployed on switch 3 and switch 1, respectively. Then, we perform routing and complete the deployment of SFC. At the next moment, new user 2 enters the network on switch 5, as shown in Figure 1b, and proposes network service requests for VNF1 and VNF3. We can migrate VNF1 to switch 5, shared by user 1 and user 2, and deploy a new VNF3 on switch 6 at the same time. In this way, the consumption of routing paths can be minimized while sharing network load and fulfilling service requests.

Therefore, to complete the dynamic deployment of SFC, we first need to predict the unknown network traffic and then perform the subsequent VNF adjustment operation.

### 2.2. Consumption of Time and Resources

After the network traffic prediction is completed, if the VNF resources are always reserved based on the highest traffic peak, it will cause a serious waste of resources. Therefore, we need to dynamically adjust the VNF resources according to the predicted results. When the predicted traffic rises, add more virtual machines (VMs) to expand the VNFs. When the predicted traffic drops, delete VMs to reduce VNFs. Next, we analyze the operation consumption from two aspects, time and resources. Table 1 shows the time consumed by different operations on the VM, where min represents minutes and s represents seconds.

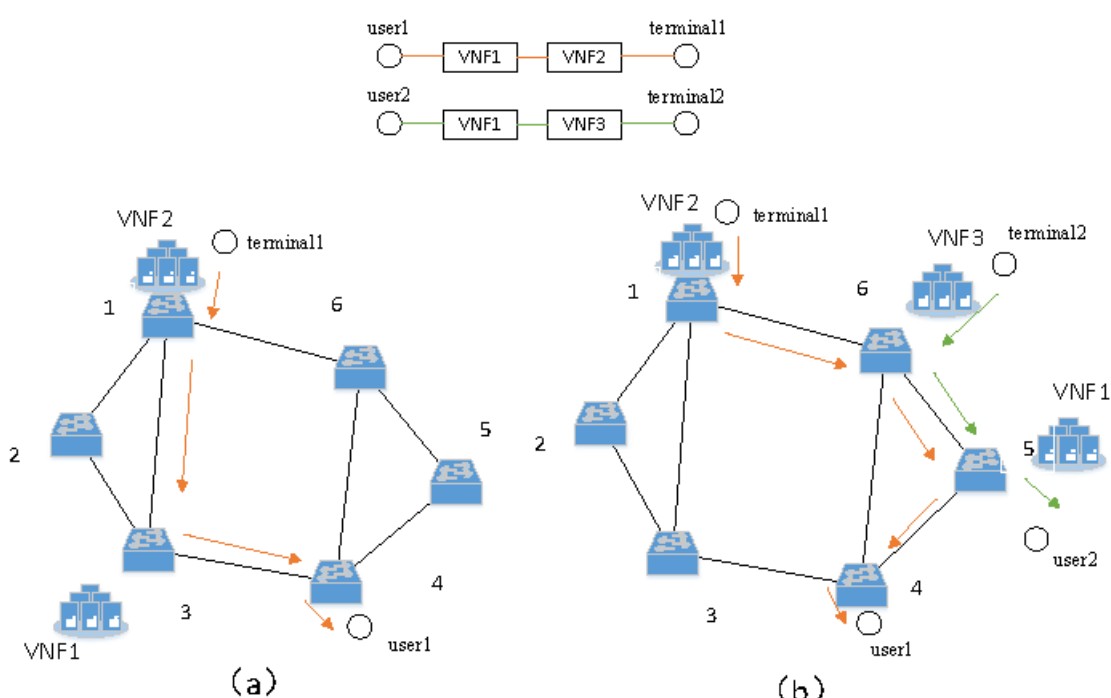

**Figure 1.** Dynamic deployment of SFC. (**a**) One user is connected to the network; (**b**) Another user is also connected to the network.

**Table 1.** The time consumed by different VM operations [4].

| Different Operations of the VM | Time Spent |
| --- | --- |
| Create new VM | 6 min |
| VM deletion | 5 s |
| VM migration | 3 min |
| traffic migration | 2 s |

As can be seen from the above table, it takes a lot of time to create a new VM, about 6 min. When there is a surge in network traffic, there are two ways to deal with it:

One is by buffering excess traffic (within cache tolerance) for a long time to wait for a new VM to be established. However, this will add a lot of operational overhead to the network operator. The other is by denying service to some network traffic, which will greatly reduce the quality of service (QoS) of the network. Compared with the newly created VM, it only takes 3 min to migrate the VM, which is relatively time-saving. However, the migration of the VM will consume a lot of network resources [4], which is not considered in this paper, and it only takes 5 s to delete the VM. This operation can save a lot of network resources. Therefore, when traffic drops, redundant VMs can be deleted to save resources. Finally, it can be seen that it only takes 2 s to migrate traffic to other paths. For online services, a delay of 2 s does not have much impact on users. Therefore, it is also a good method to integrate traffic migration into paths with fewer VMs, which can release more VMs and save more resources. We will study this method in the future. Since OpenNF supports lossless and data-preserving flow state migration [16], this paper does not consider the specific flow state migration process, but mainly considers the creation and deletion of VMs.

In addition, the operation of VNF needs to consume network resources and generate certain overheads. The overhead of deploying a new VNF is much higher than the overhead of keeping a VNF running [6]. Therefore, when designing a VNF scaling method, it is not only necessary to consider minimizing newly deployed VNFs to reduce the deploying

overhead but also to appropriately delete redundant idle VNFs to reduce the running overhead, thereby minimizing the operator's operational overhead.

### 3. Model Building

*3.1. Traffic Prediction Model Based on Improved BSO Algorithm*

3.1.1. Improved BSO Algorithm

The BSO algorithm is an optimization algorithm that expands the individuals in the BAS into groups, that is, the combination of the PSO algorithm and the BAS algorithm. The BAS algorithm was proposed by Jiang in 2017 [17]. The algorithm is inspired by the phenomenon that long-horned beetles perceive the smell of food in the environment through two antennae in nature, and keep moving in the direction of the stronger smell until they find the food.

The BAS algorithm enables the PSO algorithm to have the ability to perceive the surrounding environment and expand the range of particle information sources [15]. Compared with the BAS algorithm, the BSO algorithm is better at dealing with high-dimensional functions. It is not limited by the initial position of the beetle, which makes the BSO algorithm have a better optimization effect and efficiency [11]. Inspired by the idea of the PSO algorithm, we express the mathematical model of BSO as follows: First, $n$ beetles of the population in the $L$-dimensional search space are expressed as $X = (x_1, x_2, \cdots, x_n)$. The position of the $i$th beetle in the population in the $L$-dimensional search space is expressed as $X_i = (x_{i1}, x_{i2}, \cdots, x_{in})^T$. The velocity is expressed as $V_i = (v_{i1}, v_{i2}, \cdots, v_{iL})^T$. The individual extreme value of the population is expressed as $G_i = (g_{i1}, g_{i2}, \cdots, g_{iL})$. The population extreme value is expressed as $Q_i = (q_{i1}, q_{i2}, \cdots, q_{iL})$.

$$x_{il}^{m+1} = x_{il}^m + \alpha v_{il}^m + (1 - \alpha)\xi_{il}^m \tag{1}$$

$$v_{il}^{m+1} = \omega v_{il}^m + c_1 r_1 (g_{il}^m - x_{il}^m) + c_2 r_2 (q_{il}^m - x_{il}^m) \tag{2}$$

Equation (1) is the position change of the beetles, where $i = 1, 2, \cdots, n, l = 1, 2, \cdots, L$, $m$ is the current iteration number. $\alpha$ is a normal number, $\xi_{il}$ represents the displacement of the beetles and $v_{il}$ is the velocity of the beetles. Equation (2) represents the speed of the beetles. $c_1, c_2$ are the learning factor representing any normal number. $r_1$ and $r_2$ represent the constant between $(0, 1)$. $\omega$ represents the inertia weight.

$$\omega = \omega_{\max} - \frac{\omega_{\max} - \omega_{\min}}{M} \cdot m \tag{3}$$

Equation (3) represents the update method of the inertia weight, where $\omega_{\max}$ and $\omega_{\min}$ are the maximum and minimum weights. $m$ is the current iteration number. $M$ is the maximum iteration number. The larger the inertia weight, the stronger the global search ability. The smaller the inertia weight, the stronger the local search ability. In this paper, the inertia weight decreases as the number of iterations increases, so that the algorithm can search a larger area at the beginning to determine the range of the optimal solution. Then, as the number of iterations increases, the inertia weight becomes smaller, and the local search ability becomes stronger. The optimal solution is found within the optimal range to improve the convergence effect.

$$\zeta_{il}^{m+1} = \delta^m \cdot v_{il}^m \cdot sign(f(x_{rl}^m) - f(x_{ll}^m)) \tag{4}$$

$$x_{ll}^{m+1} = x_{ll}^m - v_{il}^m \cdot d/2 \tag{5}$$

$$x_{rl}^{m+1} = x_{rl}^m + v_{il}^m \cdot d/2 \tag{6}$$

Equation (4) is the displacement of the beetle, where $\delta$ is the step length. Equations (5) and (6) respectively represent the search behavior of the left and right antennae.

$$\delta^{t+1} = eta \cdot \delta^t \tag{7}$$

$$eta = e^{-\pi \cdot m / M} \tag{8}$$

The improvement of the original BSO method in this section lies in Equation (7), which is the improved method to calculate the step size. To improve the convergence speed and effect at the same time, a normal distribution function is introduced as the step size adjustment factor *eta*. As shown in Equation (8), this function decreases slowly in the early stages, which is conducive to speeding up the convergence speed. It decreases rapidly in the later stages, which can improve the accuracy of optimization.

$$d^t = \delta^t / c \tag{9}$$

Equation (9) is the correspondence between the search distance and the step size where $c$ is the impact factor, which can be changed manually.

3.1.2. Traffic Forecast Model

In this paper, we use $\{x_i | i = 1, 2, \cdots, n\}$ to represent the network traffic time series. Firstly, a new time series can be obtained by the phase space reconstruction method: $X(i) = \left\{ x_{i-(l-1)\tau}, \cdots, x_{i-\tau}, x_i \right\}$, where $\tau$ is the delay time and $l$ is the search dimension. Secondly, when dealing with the problem of network traffic prediction, a large number of additional parameters are involved. The high randomness of the problem makes the prediction method unable to dynamically adjust to the changing solution space.

In this regard, this paper adopts a network traffic prediction model based on the SVM algorithm. In addition, the network traffic data is reconstructed in combination with chaos theory to improve the optimization level of the algorithm and improve the prediction accuracy. However, when the SVM algorithm is used in nonlinear environments such as network traffic prediction, the problem of SVM regression arises, that is, the regression field is too large and the prediction is inaccurate. Usually, the linear regression equation $f(x) = \omega \cdot x + b$ can be used to fit the SVM [10] to solve the SVM regression problem, where $\omega$ is the weight vector and $b$ is the bias vector. This paper adopts the principle of structural risk minimization to optimize the SVM regression function as:

$$\min J = \frac{1}{2} \|\omega\| + C \cdot \sum_{i=1}^{n} (\zeta_i^* + \zeta_i) \tag{10}$$

$$\begin{cases} \zeta_i^*, \zeta_i \geq 0 \\ y_i - \omega \cdot \varphi(x) - b \leq \varepsilon + \zeta_i \\ \omega \cdot \varphi(x) + b - y_i \leq \varepsilon + \zeta_i^* \end{cases} \tag{11}$$

As shown in Equations (10) and (11), where $\|\omega\|$ is related to the complexity of the regression function. $\varepsilon$ is the insensitive loss function. $\zeta_i, \zeta_i^*$ are the relaxation factors. The penalty factor is expressed by $C$ and is closely related to the accuracy of traffic prediction. If the penalty factor is too small, the training error will be large, resulting in a weak generalization ability of the algorithm and a high prediction error. If the penalty factor is too large, the learning accuracy of the algorithm will be low, and the generalization ability of the algorithm will be weak. To improve the solution efficiency, the Equations (10) and (11) are transformed into the dual form:

$$f(x) = \sum_{i=1}^{n} (\alpha_i - \alpha_i^*) \cdot (\varphi(x_i), \varphi(x)) + b \tag{12}$$

For nonlinear prediction problems such as traffic prediction, the vector inner product $(\varphi(x_i), \varphi(x))$ in the high-dimensional space is replaced by the kernel function $k(x_i, x)$ to avoid the influence of dimension. The regression function can be expressed as:

$$f(x) = \sum_{i=1}^{n} (\alpha_i - \alpha_i^*) \cdot k(x_i, x) + b \tag{13}$$

In this paper, $K(x_i, x_j) = \exp\left(-\|x_i - x_j\|^2 / (2\sigma^2)\right)$ is used as the radial basis sum function of the SVM function, where $\sigma$ is the width parameter of the radial basis kernel function.

To improve the service quality of the network, it should be ensured that there are sufficient resources to handle the traffic at each moment [18]. Therefore, we not only need to improve the accuracy of predicted traffic but also need to estimate the upper limit of traffic. We use $x(i)$ to represent the time series of network traffic, then the predicted traffic at time $t$ can be expressed as:

$$B(T) = \text{minmax} x(t) \tag{14}$$

where $x(t)$ is the estimate based on previous predictions of traffic by using SVM.

### 3.2. VNF Deployment Model

3.2.1. Network Model

Abstract the physical network as a weighted undirected graph $G(V, L)$, where $V$ is the set of nodes and $L$ is the set of links. Each node represents a computing node. For any node $n \in N$, $C_n$ represents the remaining computing resources of the node. Each node can deploy one or more VMs, and each VM can only instantiate one VNF. The VNF is represented by the set $V = \{1, 2, \ldots, v\}$. $c_v$ represents the computing resources consumed by instantiating the VNF. $tt_v$ represents the throughput of the VNF. Each VNF can be set to active and idle states. The set of VNFs in the active state at a time $T$ is denoted by $H_v(T)$, where $h_v(T)$ is the number of VNFs. The set of VNFs in idle state is denoted by $K_v(T)$, and the number of VNFs is denoted by $k_v(T)$. $d_v$ is the delay generated after instantiating the VNF. For any physical link $L_{a,b} \in L$, $B_{a,b}$ represents the remaining bandwidth resources, and $D_{a,b}$ represents the link delay.

3.2.2. SFC Request Model

The SFC request is a directed graph $G_v = \left(I_g, O_g, D_g, T_g, B_g, V_g, L_g\right)$. The set of SFCs is represented as $S = \{s_g | g = 1, 2, \ldots\}$, where $I_g$ and $O_g$ represent the inflow and outflow nodes of the $g$th SFC, respectively. For any $S_g$, after the physical nodes of $I_g$ and $O_g$ are determined, $D_g$ represents the maximum delay limit of SFC. $T_g$ is the residence time of the SFC. $B_g$ is the traffic bandwidth required by the SFC. $V_g = \{f_i | i = 1, 2, \ldots\}$ represents the set of all VNFs in $S_g$. $s_{vg}$ represents whether the $v$th VNF is in the $g$th SFC that the traffic passes through. If $s_{vg} = 1$, it means it is. If $s_{vg} = 0$, it is not. $L_g$ represents the route set of $S_g$, where $L_g = (l_{0,1}, l_{1,2}, \ldots, l_{i-1,i})$ uses $l_{i-1,i}$ to represent the route from the $i - 1$th to the $i$th node, where 0 is the source point and $i$ is the destination point. If the $k$th VNF in $S_g$ is successfully deployed on the $i$th node of the underlying network, set $\mu_{k,i} = 1$, otherwise $\mu_{k,i} = 0$. For links, if the virtual link $l_{i-1,i}$ is successfully mapped to the physical link $L_{a,b}$, set $\gamma_{a,b}^{i-1,i} = 1$, otherwise $\gamma_{a,b}^{i-1,i} = 0$.

### 3.3. Target Optimization

Firstly, the main goal of this paper is to minimize the number of VNF instances. The objective function can be set as:

$$\min \sum_i f_i \tag{15}$$

At the same time, minimizing the cost of adaptive VNF scaling is also an optimization goal of this paper. In daily life, operators need to create or delete VNF instances according to changing network requests to reduce operational overhead. This article sets the VNF instance that is providing network services to the active state, and the other instances to the idle state. As the network traffic changes, the state of the VNF instance is dynamically switched to provide service for the network adaptively. The overhead in the adaptive change process can be divided into the following two aspects:

The first is the running overhead of VNF instances. In addition to the running overhead of active VNFs, this part also has additional overheads generated by VNFs that maintain an idle state, which mainly includes energy consumption overhead and virtual resource

occupation overhead [6]. Set $\gamma_v$ as the running overhead coefficient of the VNF instance for a period of time. The running overhead of the VNF instance during the whole adaptive scaling process can be expressed as:

$$YX_v = \sum_{t \in T} \sum_{f_i \in V_g} \gamma_v \cdot [k_v(t) + g_v(t)] \tag{16}$$

where $k_v(t)$ represents the number of idle VNFs at the time and $g_v(t)$ represents the number of working VNFs at the time $t$.

The second is the deployment cost of the new VNF instance. Based on the deployment of the original VNF instance, due to the increase in traffic, the deployment of the new VNF instance needs to be performed. At this time, the VM needs to perform image transfer, which will cause a lot of delays. As shown in Table 1, it will consume a lot of resources and generate a lot of deployment overhead. Set $\delta_v$ as the deployment overhead coefficient of the new VNF instance at the time $t$. The new deployment overhead of the VNF instance during the whole adaptive scaling process can be expressed as:

$$BS_v = \sum_{t \in T} \sum_{f_i \in V_g} \delta_v \cdot d_v(t) \tag{17}$$

where $d_v(t)$ represents the number of newly deployed VNF instances at the time $t$. The overhead of minimizing VNF adaptive scaling can be expressed as:

$$\min(YX_v + BS_v) \tag{18}$$

The deployment constraints of SFC are:

$$0 \leq \sum_{n \in N} \mu_{k,i} \leq 1 \tag{19}$$

$$\sum_{l_{i,i-1} \in L_g} \gamma_{a,b}^{i-1,i} \geq 1 \tag{20}$$

$$\sum_{L_{a,b} \in L} \gamma_{a,b}^{i-1,i} \leq 1 \tag{21}$$

$$\sum_{s_g \in S} \sum_{f_i \in V_g} s_{vg} \cdot B_g \leq tt_v \tag{22}$$

$$\sum_{n \in N} \sum_{f_i \in V_g} \mu_{k,i} \cdot c_v \leq 90\% \cdot C_n \tag{23}$$

$$\sum_{L_{a,b} \in L} b_{i-1,i} \cdot \gamma_{a,b}^{i-1,i} \leq B_{a,b} \tag{24}$$

$$\sum_{\substack{n \in N \\ f_i \in V_g}} \mu_{k,i} \cdot d_n + \sum_{\substack{L_{a,b} \in L \\ l_{i-1,i} \in L_g}} \gamma_{a,b}^{i-1,i} \cdot D_{a,b} \leq D_g \tag{25}$$

Equation (19) ensures that a VNF in the SFC can only be mapped to one underlying node, preventing VNF segmentation. Equation (20) ensures that the virtual link can be mapped on one or more physical links. Equation (21) indicates that only one virtual link is allowed to be arranged on a physical link to prevent the occurrence of the "ping-pong effect". Equation (22) is the throughput constraint, and the traffic handled by each VNF instance should not exceed the throughput of the VNF. Equation (23) is the computational resource constraint. To achieve load balancing, it is ensured that the computing resource consumption of each node does not exceed 90% of its computing resource capacity [19]. Because the software-based VNF occupies fewer storage resources [20], this paper does not consider it, but only considers the computing resources. Equation (24) represents the link

bandwidth constraint, where $b_{i-1,i}$ represents the bandwidth requirement of the $i-1$th to the $i$th node. Equation (25) represents the delay constraint, and the left side of the inequality is the sum of node delay and link delay.

## 4. Algorithm Design

The dynamic deployment of SFC in this paper is mainly divided into four steps: traffic prediction, data processing, adaptive VNF scaling, and path configuration. We use sets $B$, $C$ and $D$ to represent the existing active VNF instance set, the existing idle VNF instance set, and the existing VNF instance set, respectively. The number of VNF instances in the set is represented by $b$, $c$, and $d$, respectively. At the time $t$, then $b(t) + c(t) = d(t)$. We express the predicted number of required VNF instances as $a$. At the time $t$ ($t \leq$ final time $T$), we use the network traffic prediction model to predict the required number of VNF instances $a(t+1)$ at the next time. Then according to the existing VNF instance numbers $b(t)$, $c(t)$ and $d(t)$, we adaptively adjust the VNF instances. Finally, we configure the path of the adjusted VNF instance according to the network status and then complete the dynamic deployment of SFC to realize the flexible configuration of virtual resources. The specific process is shown in Algorithm 1.

---

**Algorithms 1**: SFC dynamic deployment algorithm based on traffic prediction and VNF scaling.

---

Input: physical network $G(V, L)$, SFC request $G_v = (I_g, O_g, D_g, T_g, B_g, V_g, L_g)$,
      network traffic $x(t)$.
Output: SFC dynamic deployment scheme.

---

Initialization time $t$;
*for* $t \leq T$
    Use Algorithms 2 to predict network traffic;
    Use Equation (30) to process the predicted traffic data;
    According to the processed predicted traffic, use Equation (31) to estimate the number of VNF instances required at the next moment $a(t+1)$;
    *for* each VNF in each SFC
        Use Algorithms 3 to adaptively scale the VNFs;
    *end*
    Configure the path for the adjusted VNF instance to complete the SFC deployment;
    Detect the load of the VNF instance in $B(t)$;
    *if* it exceeds the rated load of the VNF
        *if* there are redundant VNF instances in $C(t)$
            Adjust the idle VNF instance with the heavy load to the active state;
        *else*
            Deploy a new VNF instance;
        *end*
    *end*
*end*

---

### 4.1. Traffic Forecast

In this paper, we use SVM to predict network traffic and optimize it with an improved BSO algorithm. The specific idea is as follows: Firstly, we input the traffic data training set. We use the SVM algorithm to calculate its corresponding fitness value and select its individual best fitness value and group best fitness value. Then we perform an iterative operation to update the inertia weight and step size. In each iteration, the search behavior, displacement, speed, position, and fitness value of each beetle are updated. We then update the individual optimal and group optimal in the beetle swarm. Finally, through continuous iteration, the optimal value of the beetle swarm is what we want. The specific process is shown in Algorithm 2.

---

**Algorithms 2**: Network traffic prediction algorithm based on improved BSO optimized SVM.

---

Input: traffic data $x(t)$
Output: individual optimal, group optimal flow data

---

Initialize population $X_i$ and population velocity $v$;
Set parameters: step size $\delta$, upper and lower speed limits $v_{\max}$, $v_{\min}$, population size *sizepop*, the maximum number of iterations $M$;
Calculate the fitness value of the traffic data $x(t)$ with the SVM training set;
Calculate the best fitness value of the individual and the best fitness value of the group;
　　*for* each iteration in the range $M$
　　　　Set the inertia weight $\omega$ according to Equation (3);
　　　　Update the step size $d$ according to Equation (9);
　　　　*for* every beetle within *sizepop*
　　　　　　Calculate the search behavior of beetles according to Equations (5) and (6);
　　　　　　Calculate the displacement of the beetle according to Equations (4);
　　　　　　Calculate the speed of the beetle according to Equation (2);
　　　　　　According to Equation (10), update the position change of the beetle;
　　　　　　Update the fitness value $fitness(x)$ of beetles;
　　　　*end*
　　　　Record the fitness value of each beetle;
　　　　*for* every beetle within *sizepop*
　　　　　　*if* $fitness(x) <$ individual optimal
　　　　　　　　Update individual optimal;
　　　　　　*end*
　　　　　　*if* $fitness(x) <$ group optimal
　　　　　　　　Update group optimal;
　　　　　　*end*
　　　　*end*
　　　　Record the optimal value of the beetle population;
　　　　Update $\delta$ with Equation (7);
　　*end*

---

In the BSO algorithm used in this paper, the inertia weight and step size decrease with the increase of the iteration number. This prevents the algorithm from converging to the target point too quickly, thus reducing the occurrence of the phenomenon that the group falls into a local optimum. At the same time, the linear combination of the search and the speed of the beetle swarm can speed up the iteration speed. This reduces local optima and improves stability when dealing with high-dimensional problems [11]. We improved the original BSO algorithm. We adjusted the calculation method of the step size to improve the convergence rate and effectiveness of the algorithm. After the traffic prediction is completed, the predicted traffic needs to be processed accordingly.

*4.2. Traffic Data Processing*

In this paper, we take the approach of minimizing the upper limit of traffic. That is, we should try to reduce occurrences where the predicted traffic is smaller than the actual traffic, to improve the availability of network services. We should also ensure that the processed traffic is not much higher than the actual traffic to reduce network resource consumption. The specific implementation method is to reserve appropriate redundant capacity. When the actual traffic is higher than the predicted traffic, we should ensure that there are enough network resources to deal with it.

There are many ways to reserve redundant traffic. The most common method is to reserve based on the $3 - \sigma$ principle. That is, we reserve enough resources to serve the network traffic in the range of $\mu + 3\sigma$, where $\mu$ is the mean value of the network traffic and $\sigma$ is the standard deviation of the network traffic. The distribution range of the network traffic rate is large enough that the value of $\sigma$ is large. This method will waste a lot of network resources, although it can provide good service for network traffic. In this regard,

we only consider reducing the reserved redundancy according to the variance of the relative prediction error to save network resource consumption.

$$e(t) = \max\left\{ \frac{x(t) - \hat{x}(t)}{x(t)}, 0 \right\} \tag{26}$$

$$\bar{e}(t) = \frac{1-\alpha}{1-\alpha^N} \sum_{i=t-N+1}^{t} \alpha^{t-i} \cdot e(i) \tag{27}$$

$$var(t) = \frac{1-\alpha}{1-\alpha^N} \sum_{i=t-N+1}^{t} \alpha^{t-i} \cdot [e(i) - \bar{e}(i)]^2 \tag{28}$$

$$\sigma(t) = \sqrt{v(t)} \tag{29}$$

Equation (26) represents the relative traffic prediction error at the time $t$, where $x(t)$ is the actual traffic at the time $t$ and $\hat{x}(t)$ is the predicted traffic at the time $t$. Equation (27) represents the average prediction error at the time $t$, where $\alpha$ is a parameter in the range of $(0,1)$. The larger the value of $\alpha$ is, the more dependent the traffic prediction is on historical data. Therefore, we take the value of $\alpha$ closer to 0. Equation (28) represents the traffic variance at the time $t$. Equation (29) is its standard deviation. We can express the upper bound of the predicted traffic at the time $t + 1$ as:

$$B_{upper}(t+1) = \hat{x}(t+1) \cdot [1 + \bar{e}(t) + 3\sigma(t)] \tag{30}$$

In this way, we can minimize the number of moments when the predicted traffic is smaller than the actual traffic, so as to cope with the changing network traffic and improve network availability. Then, according to Equation (31), the processed predicted traffic is converted into the number of demanded VNF instances at the time $t$ according to the throughput of the VNF instance:

$$a(t) = \left\lceil \frac{B_{upper}(t)}{tt_v} \right\rceil \tag{31}$$

where $a(t)$ is the predicted number of VNF instances at the time $t$. In order to ensure sufficient resources, we take the upper limit. $tt_v$ is the throughput of VNF. The throughput and computing resource requirements of different VNF instances are shown in Table 2 [21].

**Table 2.** Throughput and computing resource requirements of different VNFs.

| VNF Instance Type | Firewall | Proxy | Nat | IDS |
|---|---|---|---|---|
| $tt_v$ (Mbps) | 900 | 900 | 900 | 600 |
| Computing resources | 4 | 4 | 2 | 8 |

*4.3. Adaptive VNF Deployment Algorithm*

Firstly, we design a VNF dynamic adjustment method based on the demand data of the VNF instance at the current moment in this section. According to the required number of VNFs and the number of existing VNFs, we dynamically create or delete VNF instances to achieve the purpose of dynamically adjusting virtual resources. Then, according to the network status, we configure the path of the adjusted VNF instance to complete the dynamic deployment of the SFC.

In daily life, the dynamic creation and deletion of VNF instances will affect the operational overhead of network operators. For example, when the network traffic decreases, the required number of VNF instances will also decrease. At this time, if the original number of VNFs continues to run, it will bring additional VNFs running overhead. However, if the VNF instance is deleted immediately, and if the network traffic increases at the next moment, new VNF instances need to be created, which will result in a large amount of

VNF deployment overhead. However, if the VNF instance is not deleted in time, and if the network traffic continues to drop at the next moment, these redundant idle VNF instances will also generate a large amount of VNF running overhead. Therefore, the overhead contradiction we need to deal with is how to reduce the VNF deployment overhead while maintaining a low VNF running overhead, so as to minimize the operator's operating overhead. The specific idea of VNF adaptive scaling is as follows:

Firstly, at the time $t$, if $a(t) \geq d(t-1)$, we set all idle VNF instances into the active state, and deploy $a(t) - d(t-1)$ new VNF instances at the same time to cope with the increase in network traffic. If $a(t) < b(t-1)$, we sort the VNF instances in ascending order according to the load size, and set the first $b(t-1) - a(t)$ VNF instances in the sequence to the idle state. This is to minimize the number of SFCs that need to change the path when deleting VNFs. If $b(t-1) \leq a(t) < d(t-1)$, we sort the idle VNF instances in descending order according to the load size, and set the first $a(t) - b(t-1)$ VNF instances as active. In this way, the load of idle VNF instances can be minimized, and the number of SFCs that need to be changed during the deletion of VNFs can be reduced. Finally, in order to optimize the cost of adaptive scaling, we limit the number of VNF instances in the idle state. We use $N_{upper}$ to denote the maximum number of the idle state VNF instances allowed in the network system. If $c(t) > N_{upper}$, we delete the VNF instance with less load. The specific process is shown in Algorithm 3.

---

**Algorithms 3**: Adaptive VNF scaling algorithm.

---

Input: physical network $G(V, E)$, set $B(t), C(t), D(t), b(t), c(t), d(t)$,
   the number of VNF instances required at the next moment $a(t+1)$
Output: Set of the next moment $B(t+1), C(t+1)$

---

*if* $a(t+1) \geq d(t)$
  Newly deploy $b(t) - a(t+1)$ VNF instances and set them to active state;
  Update node remaining resources and sets $B(t), C(t), D(t)$;
*else if* $a(t+1) < b(t)$
  Sort VNF instances in ascending order according to the load;
  Set the first $b(t) - a(t+1)$ VNF instances in the sequence to idle state;
  update sets $B(t), C(t), D(t)$;
*else*
  Sort the idle VNF instances in descending order according to the load;
  Set the first $a(t+1) - b(t)$ VNF instances to active state;
  update sets $B(t), C(t), D(t)$;
*end*
*if* $c(t) > N_{upper}$
  Sort the idle VNF instances in ascending order according to the load;
  *for* the first $c(t) - N_{upper}$ VNF instances
   *if* VNF instances carry traffic
    Use the k-shortest path algorithm to recalculate the service path and migrate the traffic;
   *end*
  *end*
  Delete the first $c(t) - N_{upper}$ VNF instances;
*end*
Obtain $B(t+1), C(t+1)$;

---

Finally, under the conditions of satisfying Equations (20), (24), and (25), we use the k-shortest path algorithm to calculate the routing paths between service function instances. We map the virtual link to the underlying physical link to forward data traffic and complete the dynamic deployment of SFC.

## 5. Experimental Results and Analysis

This part is mainly divided into three experiments according to the order of SFC dynamic deployment. They are the prediction of network traffic, the processing of predicted

data, and the adaptive scaling of VNFs. We evaluate and analyze the proposed algorithm step by step.

*5.1. Network Traffic Forecast*

In this section, we evaluate and analyze the network traffic prediction algorithm based on the improved BSO-SVM. We compare it with the network traffic prediction results based on PSO-SVM through specific evaluation indicators.

5.1.1. Experimental Setup

The network traffic used in this article comes from the hourly network access traffic of a 14-day network in a certain place [22], with a total of 14 × 24 = 336 network traffic time series. Then we use it as a sample to train the improved SVM, and the kernel function in the SVM adopts the radial basis function. Then we make predictions for the network traffic on day 15. In this paper, the number of iterations of the improved BSO algorithm is set to 200. The population size is 20. The maximum and minimum values of inertia weights are set to 0.9 and 0.4, respectively. The initial step size is set to 10. In the PSO algorithm, the parameters such as iteration times, population size, and learning factor are the same as those in the BSO algorithm.

5.1.2. Evaluation Indicators

We use two parameters, mean squared error (MSE) and squared correlation coefficient (SCC), to measure the error and correlation between the predicted value and the actual value. Then we evaluate the performance of this algorithm.

$$MSE = \sum_{i=1}^{n} (x_i - \hat{x}_i)^2 / n \tag{32}$$

$$SCC = Cov^2(x, \hat{x}) / [D(x) \cdot D(\hat{x})] \tag{33}$$

As shown in Equations (32) and (33), $x$ and $x_i$ represent actual values. $\hat{x}$ and $\hat{x}_i$ represent predicted values. $n$ is the number of samples. The MSE reflects the error condition of the prediction. The smaller the value, the more accurate the traffic forecast and the better the forecast performance. The SCC reflects the correlation between the predicted value and the actual value. The higher the value, the more accurate the traffic prediction result and the better the prediction performance.

We compare the performance difference between the BSO algorithm and the PSO algorithm in traffic prediction through the fitness curve comparison chart and the result error chart.

5.1.3. Experimental Results

Table 3 shows the comparison of traffic prediction results between BSO-SVM and PSO-SVM (the improved BSO-SVM and the original BSO-SVM algorithm have the same traffic prediction results). In order to reduce the influence of random factors, we carry out 10 simulation experiments and select the average value of each result as the final result of the experiment.

**Table 3.** Comparison of experimental data with different methods.

| Method of Prediction | Training Set | | Test Set | |
|---|---|---|---|---|
| | MSE (%) | SCC (%) | MSE (%) | SCC (%) |
| BSO-SVM | 0.58 | 94.61 | 3.64 | 78.75 |
| PSO-SVM | 0.15 | 98.59 | 5.58 | 64.79 |

It can be seen from the analysis of the data in Table 3. In the test set, the MSE obtained by using the BSO-SVM method is 3.64%, which is lower than the 5.58% obtained by using the PSO-SVM method. The SCC obtained by using the BSO-SVM method was 78.75%, which was higher than the 64.79% obtained using the PSO-SVM method. It can be seen that the algorithm proposed in this paper has less error and higher accuracy when predicting network traffic.

Figure 2 is a graph of the fitness changes when PSO, BSO, and improved BSO algorithms are used to optimize SVM. The improved BSO algorithm is represented by "*BSO". It can be seen that the improved BSO algorithm has a faster convergence speed and better convergence effect than the original BSO algorithm. Compared with the PSO algorithm, the final fitness convergence value of "*BSO" is smaller, and the convergence speed and global search ability are better. Its learning accuracy is higher.

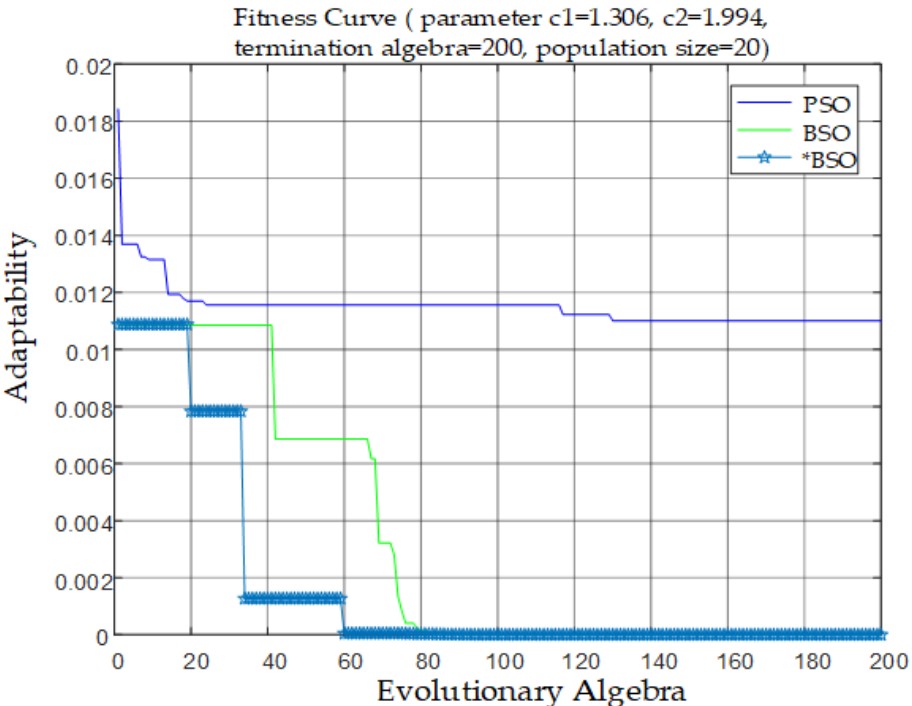

**Figure 2.** Comparison of fitness curves.

It can be seen from Figure 3b–d that the relative error of the predicted value of BSO-SVM is more stable than that of PSO-SVM. It is closer to the actual network traffic value, and the prediction is more accurate. Therefore, the network traffic prediction model based on BSO-SVM proposed in this paper has a better optimization effect and better performance.

*5.2. Traffic Data Processing*

In order to improve the availability of network services, we process the predicted network traffic data by minimizing the traffic cap. In this way, sufficient network resources are reserved for traffic without excessive consumption of network resources.

Firstly, we evaluate the effect of the proposed BSO-SVM algorithm on reserving network resources. We set $T_{under}$ to be the number of moments when the predicted traffic is lower than the actual traffic. Then we compare the two cases with and without data processing to reflect the resource reservation effect of the proposed data processing method. BSO-SVM and PSO-SVM are also compared to reflect the advantages of high accuracy of traffic prediction.



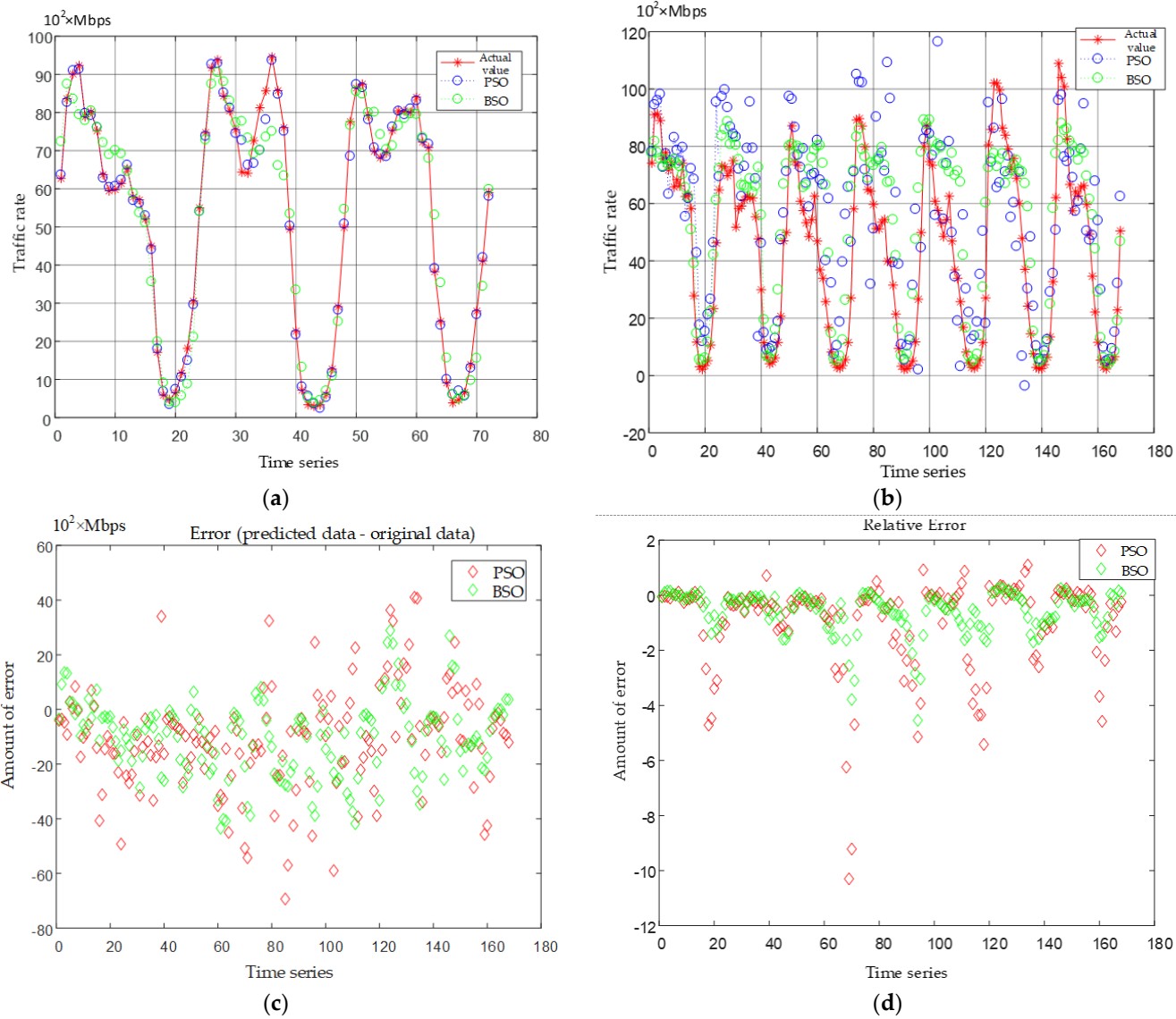

**Figure 3.** Comparison of forecasting effects with different methods. (**a**) Comparison of training results; (**b**) Comparison of predicting results; (**c**) Comparison of predicting error; (**d**) Comparison of relative error.

As shown in Figure 4, the left side is the result of using the BSO-SVM method to predict traffic, and the right side is the result of using the PSO-SVM method. The yellow is the unprocessed $T_{under}$ of the predicted data, and the purple is the $T_{under}$ after the predicted data has been processed. It can be seen that the processed $T_{under}$ of the data is much lower than the unprocessed $T_{under}$. In particular, using the BSO-SVM prediction method, the $T_{under}$ is zero after the data is processed. It can ensure that the predicted traffic at each moment in the entire traffic prediction process is higher than the actual traffic, which achieves a good resource reservation effect and is conducive to improving the availability of services. By comparing different prediction methods, it can also be seen that no matter whether the data is processed or not, the $T_{under}$ generated by BSO-SVM for prediction is less than that of PSO-SVM. Since the prediction of BSO-SVM is more accurate, its relative error is more stable, which is more conducive to data processing. Therefore, it is beneficial to reserve virtual resources and improve service availability. In addition, the $T_{under}$ generated by the processing method of $3 - \sigma$ is also zero. However, this processing method will waste a lot of network resources, as shown in the following figure.

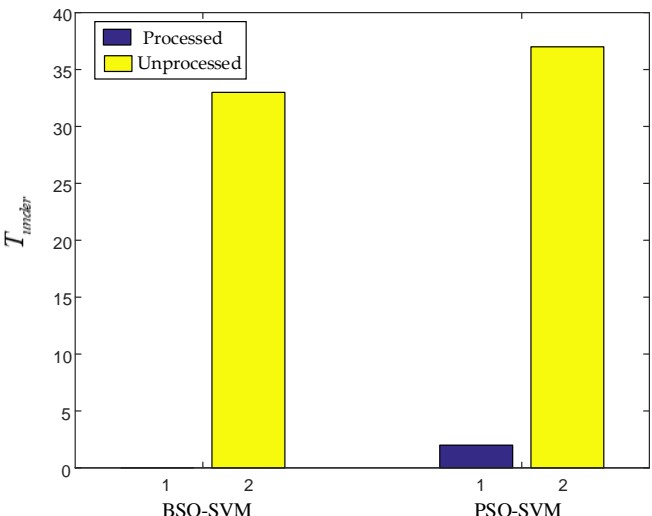

**Figure 4.** Comparison of the results $T_{under}$.

Figure 5 shows the resource consumption of different processing methods, where the resource consumption is represented by the sum of the reserved bandwidth resources at each moment. It can be seen that the bandwidth resources reserved by using the $3 - \sigma$ processing method are much higher than the processing method in this paper and the actual requirements. In the case of the same $T_{under}$, the processing method of $3 - \sigma$ will waste more virtual network resources and add more running overhead for the network operator. The processing method proposed in this paper reserves appropriate network resources, and the difference is small compared to the actually required network resources. While improving service availability, network resource consumption is reduced. By comparing the resource consumption of the $3 - \sigma$ processing method with the processing method proposed in this paper, it can be seen that the more accurate the traffic prediction is, the more resources can be saved. The reason for the high prediction accuracy of the processing method proposed in this paper is that it uses the $3 - \sigma$ principle only on the error of traffic prediction, rather than the entire traffic. In this way, it can better change with the change of traffic variance, so as to better fit the real traffic.

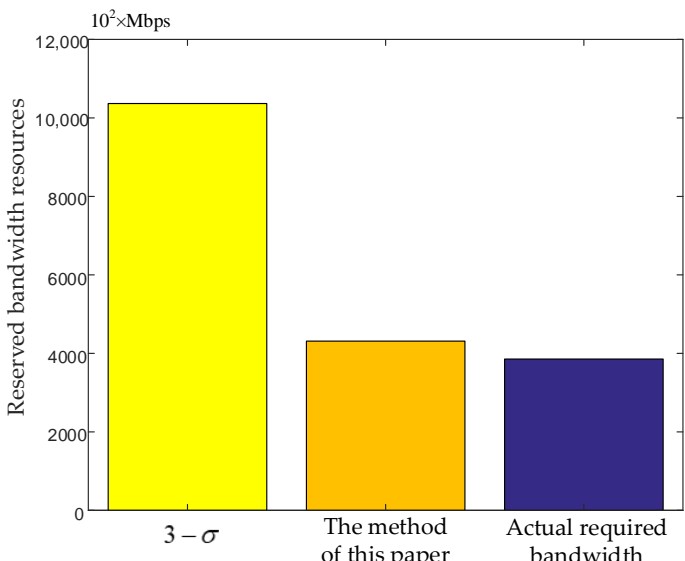

**Figure 5.** Comparison of resource consumption with different processing methods.

The blue and red lines in Figure 6 represent the actual traffic and the predicted traffic after processing, respectively. It can be seen that the predicted traffic after processing is

always higher than the actual traffic, which achieves a good resource reservation effect and is beneficial to the improvement of service availability.

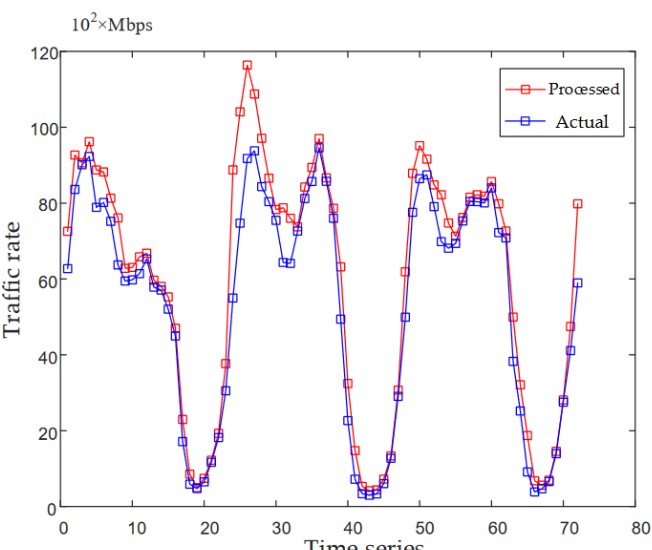

**Figure 6.** Traffic comparison chart.

*5.3. Dynamic VNF Deployment*

In this section, we evaluate and analyze the proposed VNF dynamic deployment method. Firstly, we convert the processed predicted traffic into the required number of VNF instances by Equation (31). Then we adaptively scale the VNF according to the number of demands to achieve dynamic adjustment of virtual resources.

We set the dynamic adaptive VNF scaling method proposed in this paper as method 1. Set the VNF scaling method that does not limit the number of idle VNFs as method 2. Set the VNF scaling method that does not set the idle state of the VNF and changes with the change in VNF demand as method 3. Set the VNF scaling method based on the BSO-SVM algorithm to predict traffic without data processing as method 4. Set the VNF scaling method based on the PSO-SVM algorithm to predict traffic without data processing as method 5. The specific comparison of the five methods is shown in Table 4. Then we compare and analyze these five methods with the operating overhead as the evaluation index.

**Table 4.** Comparison of different scaling methods.

| VNF Scaling Method | Traffic Forecast Method | Whether It Has Undergone Data Processing | How to Delete VNF |
|---|---|---|---|
| Method 1 | BSO-SVM | √ | more than $N_{upper}$ |
| Method 2 | BSO-SVM | √ | not delete |
| Method 3 | BSO-SVM | √ | delete immediately |
| Method 4 | BSO-SVM | × | more than $N_{upper}$ |
| Method 5 | PSO-SVM | × | more than $N_{upper}$ |

5.3.1. Experimental Setup

The physical network is a connectivity graph composed of 100 nodes and 525 links. It is a medium-sized network topology graph. We set the sequence of VNFs in the SFC request to Firewall → IDS → Nat → Proxy. Its throughput and computing resource requirements are shown in Table 2. Set the running overhead coefficient $\gamma_v$ of the VNF instance to 1, the deployment overhead coefficient $\delta_v$ to 5, and the maximum number of idle VNF instances $N_{upper}$ to 2.

### 5.3.2. Experimental Results

As shown in Figure 7, the green line represents the required number of VNF instances (firewalls) converted from the processed predicted traffic. It can be seen that the changing trend of the number of firewall instances is roughly the same as the changing trend of the traffic. The number of instances changes with the dynamic changes of network traffic, which can meet the dynamic virtual resource requirements. This paper sets up active and idle VNF instances to optimize resource allocation. When the network is working normally, the active VNF instances provide network services to users. When the traffic increases, the number of VNF instances in the idle state decreases, and the idle state turns into an active state to provide network services to users. When the traffic decreases, the number of required VNFs decreases and the number of instances in the idle state increases accordingly (the number is limited by $N_{upper}$). In this way, we can not only reduce the number of newly deployed VNF instances due to traffic surges but also avoid the continuous creation and deletion of VNF instances due to the small fluctuations in traffic. Thus, the deployment overhead of network operators is reduced.

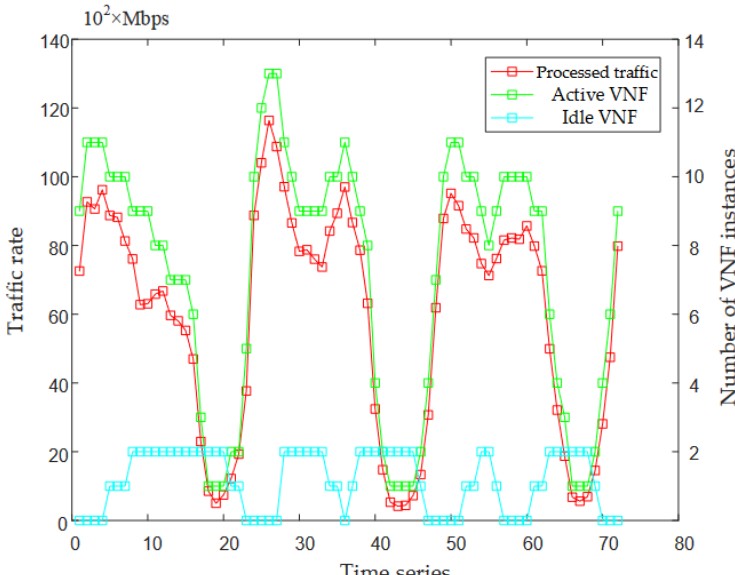

**Figure 7.** The graph of network traffic and the number of VNFs in different states.

As shown in Figure 8, purple, green, and yellow are, respectively, the network overheads of Method 1, Method 2, and Method 3. Firstly, we analyze the deployment overhead. Method 1 dynamically creates and deletes VNFs according to dynamically changing VNF requirements. In method 2, no deletion is performed, but a new VNF is created when the traffic increases, resulting in a small deployment overhead. Therefore, the deployment overhead of method 1 is much higher than that of method 2. Method 3 does not set the VNF instance to the idle state, which makes it more susceptible to dynamic changes in traffic. We need to create and delete VNF instances more frequently, which makes the deployment overhead higher than method 1.

Secondly, we analyze the running overhead. Method 2 does not delete the VNF. When the traffic drops, it maintains the running of many idle VNF instances. Method 1 performs a certain number of deletion operations on the VNF and only maintains a part of the idle state of the VNF instance to run, resulting in a relatively small running overhead. Therefore, the running overhead generated by method 2 is much higher than that generated by method 1. Method 3 does not set idle VNF instances and does not need to maintain the running of idle VNF instances. Only the active VNF instances that are working will generate running overhead. However, method 1 still needs to maintain the work of a certain number of idle VNF instances. Therefore, the running overhead generated by method 1 is higher than that generated by method 3.

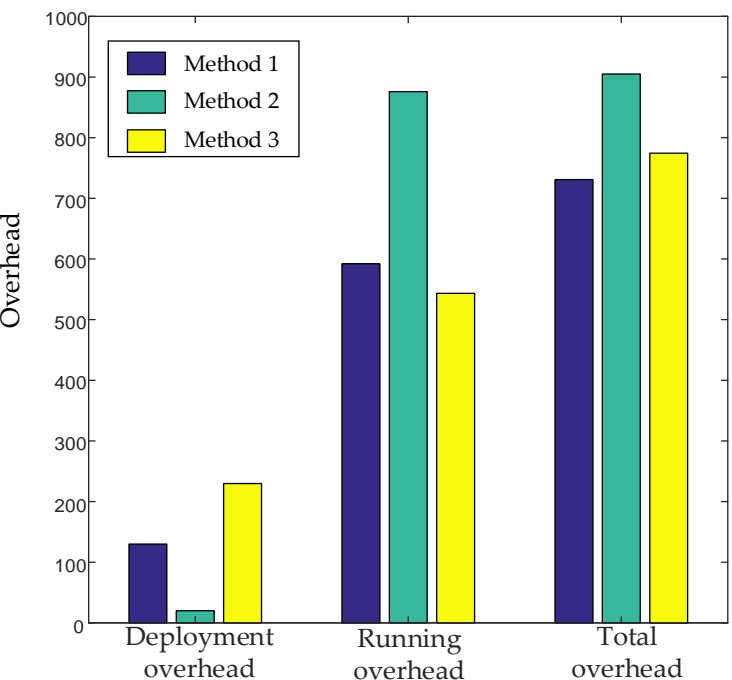

**Figure 8.** Comparison of network overhead among method 1, method 2, and method 3.

Finally, we analyze the total cost. It can be seen from the figure that method 1 has the smallest total cost compared with the other two methods. Because when the operator provides network services, the deployment overhead of creating a new VNF instance is much higher than the running overhead of maintaining the operation of the VNF instance [6]. Therefore, in order to reduce the total overhead of network operators, the number of newly-created VNF instances should be minimized.

As shown in Figure 9, the pink is the overhead generated by method 4, and the gray is the overhead generated by method 5. It can be seen that the deployment overhead of method 5 is higher than that of method 4. The running overhead of method 5 is the same as that of method 4. The total overhead of method 5 is higher than that of method 4. The difference between method 4 and method 5 is that the accuracy of the traffic forecasting method we use is different. The accuracy of the traffic prediction method based on BSO-SVM adopted by method 4 is higher than that of the traffic prediction method based on PSO-SVM adopted by method 5, so the overhead of method 4 is lower than that of method 5. Our analysis shows that due to the relatively low accuracy of PSO-SVM, the predicted traffic results fluctuate greatly. The creation and deletion of VNF instances need to be performed more frequently, resulting in more deployment overhead, thereby increasing the total overhead. From this, it can be concluded that the higher the accuracy of the predicted traffic is, the more beneficial it is to reduce the total operating overhead.

As shown in Figure 10, the purple is the overhead generated by method 1, and the pink is the overhead generated by method 4. It can be seen that the deployment overhead and running overhead of method 1 are slightly higher than those of method 4. The reason is that in the first method, the traffic prediction data is processed. That is, in order to improve the availability of network services, we add an appropriate amount of redundancy to the predicted network traffic, which results in extra overhead. However, it can be seen from Figure 4 that after data processing, the value of $T_{under}$ is significantly reduced, which greatly improves the availability of network services. Therefore, it is worthwhile for method 1 to significantly improve the availability of network services at the cost of a small amount of overhead.

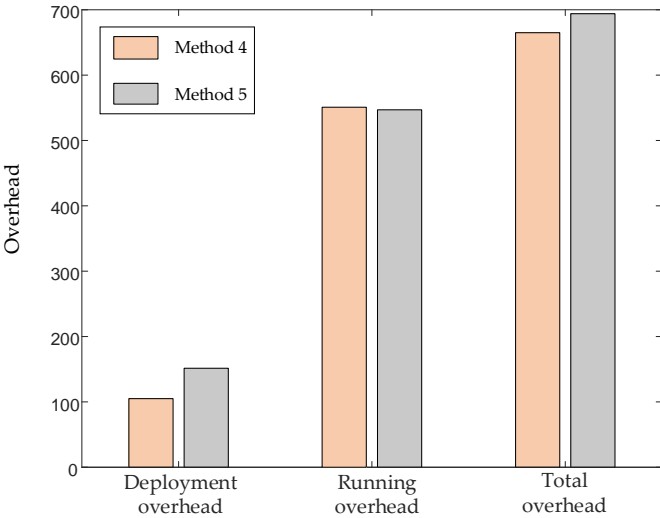

**Figure 9.** Comparison of network overhead between method 4 and method 5.

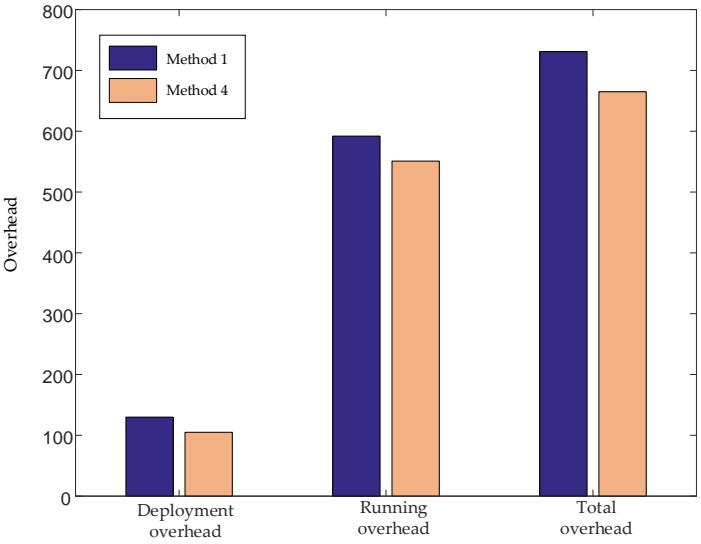

**Figure 10.** Comparison of network overhead between method 1 and method 4.

## 6. Conclusions

This paper studies the deployment of SFC in the scenario of dynamic traffic changes. Firstly, for the problem of network traffic forecasting, we propose a network traffic forecasting model based on the improved BSO algorithm to optimize the SVM. In this way, the prediction accuracy of network traffic can be improved, so as to prepare for the subsequent improvement in service availability and the reduction in network operator overhead. Secondly, in order to apply the predicted traffic to the dynamic scaling of VNF, we add appropriate redundancy to the predicted traffic data to maximize the network services that can be provided while saving network resources. Then, based on the processed traffic data, we designed a VNF scaling method, which can flexibly create and delete VNF instances according to the dynamically changing network traffic. This enables the dynamic management of virtual resources and achieves the purpose of saving overhead. Finally, we use the k-shortest path algorithm to calculate the routing paths between VNF instances to complete the dynamic deployment of SFC. The experimental results show that the method proposed in this paper can effectively improve the accuracy of traffic prediction and improve the availability of network services. Ultimately, it can reduce network resource consumption and operational overhead. It has had a good optimization effect. In the future, we will

study the dynamic routing method based on traffic migration to further realize the dynamic deployment of SFC.

**Author Contributions:** Conceptualization and methodology, H.H. and Q.K.; software, H.H. and S.Z.; validation, H.H. and J.W.; writing—original draft preparation, H.H.; writing—review and editing, H.H., Q.K. and Y.F. All authors have read and agreed to the published version of the manuscript.

**Funding:** This research received no external funding.

**Data Availability Statement:** Not applicable.

**Conflicts of Interest:** The authors declare no conflict of interest.

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
