# Peer review of "Service Function Chain Deployment Method Based on Traffic Prediction and Adaptive Virtual Network Function Scaling"

_electronics, doi:10.3390/electronics11162625_

Round 1

Reviewer 1 Report

This paper considers the deployment of SFC in the scenario of dynamic traffic changes. The authors proposed an SFC deployment method based on traffic prediction and adaptive virtual network function (VNF) scaling.

The originality of the proposed method is that it is divided into three steps: network traffic prediction, data processing and adaptive VNF scaling. Furthermore, the numerical results show that the proposed method caneffectively improve the availability of network services and reduce the operating overhead.

Overall, the paper is well written and well presented, and the paper could be improved by considering the following:

1) spell and grammar mistakes

2) Check equations 32 and 33. e.g. for equation 32, shouldn't it be  (x - xi) instead of (xi - xi). 

3) As future works, it would be interesting to integrate traffic migration into paths with fewer VMs, which can release more VMs and save more resources.

4) in the numerical results section (line 461), it is stated that 10 simulation experiments were carried out. Is 10 experiments sufficient? if yes, please justify why 10 experiments are sufficient.

5) Add legends to graphs of figures 4, 8, 9 and 10. Otherwise, it is not clear to distinguish which result corresponds to which method.

Reviewer 2 Report

The paper is well written and present a very interesting topic. I have the following minor recommendation and concerns.

1- The process of deploying SFC requires (i) VNF placement, (ii) chaining, and (iii) scheduling, before tackling the (iv) migration. How does authors go through this process to reach the migration.

2- As the number of VNF augment on a given infrastructure, how would the proposed solution cope with that? As a solution, in the literature, mean-field type games are proposed to deal with such a problem. Refer to the work in https://doi.org/10.1109/JSYST.2022.3171232

3- I strongly suggest authors to investigate the use of containers instead of VMs as the solution is being largely adopted now.

4- As for the presentation level, I strongly recommend authors to rephrase some of the sentences.

5- I recommend the use of algorithms instead of tables (table 2, 3, and 5).

6- Figure 8 to 10, the y-axis label is missing.
